

# Molecular and agro-morphological diversity assessment of some bread wheat genotypes and their crosses for drought tolerance

Mohamed A. Ezzat[1], Nahaa M. Alotaibi[2], Said S. Soliman[1], Mahasin Sultan[1], Mohamed M. Kamara[3], Diaa Abd El-Moneim[4], Wessam F. Felemban[5,6], Nora M. Al Aboud[7], Maha Aljabri[7], Imen Ben Abdelmalek[8], Elsayed Mansour[9] and Abdallah A. Hassanin[1]

[1] Genetics Department, Faculty of Agriculture, Zagazig University, Zagazig, Egypt
[2] Department of Biology, College of Science, Princess Nourah bint Abdulrahman University, Riyadh, Saudi Arabia
[3] Department of Agronomy, Faculty of Agriculture, Kafrelsheikh University, Kafr El-Sheikh, Egypt
[4] Department of Plant Production, (Genetic Branch), Faculty of Environmental and Agricultural Sciences, Arish University, El-Arish, Egypt
[5] Biological Department, Faculty of Science, King Abdulaziz University, Jeddah, Saudi Arabia
[6] Immunology Unit, King Fahd Medical Research Center, King Abdulaziz University, Jeddah, Saudi Arabia
[7] Department of Biology, Faculty of Science, Umm Al-Qura University, Makkah, Saudi Arabia
[8] Department of Biology, College of Science, Qassim University, Saudi Arabia
[9] Department of Crop Science, Faculty of Agriculture, Zagazig University, Zagazig, Egypt

Corresponding authors
Elsayed Mansour,
sayed_mansour_84@yahoo.es
Abdallah A. Hassanin,
asafan@zu.edu.eg

## ABSTRACT

Wheat, a staple cereal crop, faces challenges due to climate change and increasing global population. Maintaining genetic diversity is vital for developing drought-tolerant cultivars. This study evaluated the genetic diversity and drought response of five wheat cultivars and their corresponding F1 hybrids under well-watered and drought stress conditions. Molecular profiling using ISSR and SCoT-PCR markers revealed 28 polymorphic loci out of 76 amplified. A statistically significant impact of parental genotypes and their crosses was observed on all investigated agro-morphological traits, including root length, root weight, shoot length, shoot weight, proline content, spikelet number/spike, spike length, grain number/spike, and grain weight/spike. The parental genotypes P1 and P3 had desirable positive and significant general combining ability (GCA) effects for shoot fresh weight, shoot dry weight, root fresh weight, root dry weight, shoot length, and root length under well-watered conditions, while P3 and P5 recorded the highest GCA estimates under drought stress. P3 and P4 showed the highest GCA effects for number of spikelets per spike, the number of grains per spike, and grain weight per spike under normal conditions. P5 presented the maximum GCA effects and proved to be the best combiner under drought stress conditions. The cross P1× P3 showed the highest positive specific combining ability (SCA) effects for shoot fresh weight under normal conditions, while P2×P3 excelled under water deficit conditions. P1× P2, P1 × P3, and P4× P5 were most effective for shoot dry weight under normal conditions, whereas P1×P3 and P3×P5 showed significant SCA effects under drought stress. Positive SCA effects for root fresh weight and shoot length were observed for P3×P5 under stressed conditions. Additionally, P4×P5 consistently recorded the

highest SCA for root length in both environments, and P3×P5 excelled in the number of spikelets, grains per spike, and grain weight per spike under drought conditions. The evaluated genotypes were categorized based on their agronomic performance under drought stress into distinct groups ranging from drought-tolerant genotypes (group A) to drought-sensitive ones (group C). The genotypes P5, P2×P5, and P3×P5 were identified as promising genotypes to improve agronomic performance under water deficit conditions. The results demonstrated genetic variations for drought tolerance and highlighted the potential of ISSR and SCoT markers in wheat breeding programs for developing drought-tolerant cultivars.

## INTRODUCTION

Bread wheat (*Triticum aestivum* L.) is a staple food crop crucial for global food security and faces significant challenges due to increasing drought occurrences exacerbated by climate change (*Chauhdary et al., 2024*). Drought stress impairs wheat growth and development, leading to reduced growth, productivity and grain quality (*Galal et al., 2023*). This environmental stress affects critical physiological processes, including photosynthesis, nutrient uptake, and water regulation (*Habibullah et al., 2021*). The severity of drought impact on wheat is further compounded by the crop vulnerability to fluctuating water availability during key growth stages, such as flowering and grain filling (*Mannan et al., 2022*). Addressing these challenges requires a multifaceted approach that includes developing drought-resistant wheat varieties, improving agronomic practices, and implementing effective water management strategies to sustain wheat production in the face of growing climatic uncertainties (*Kamara et al., 2022*).

Climate change is significantly intensifying drought stress globally, posing a formidable challenge to ecosystems, water resources, and agricultural production (*Bas & Killi, 2024*; *Rezaei et al., 2023*). Rising temperatures elevate evaporation rates, increasing water demand while simultaneously diminishing its availability. Unpredictable weather patterns result in more frequent and prolonged drought periods, exacerbating water scarcity (*Chaudhry & Sidhu, 2022*; *Grigorieva, Livenets & Stelmakh, 2023*). This intensification of drought stress threatens freshwater resources and severely impacts agricultural production(*Fawzy et al., 2020*; *Rosa, 2022*). In light of these challenges, assessing wheat genetic resources for future utilization is of paramount importance (*Guzzon et al., 2022*; *Pequeno et al., 2024*). Moreover, integrating pre-breeding materials and existing cultivars into genomics-assisted breeding programs offers immense potential for improving the productivity of wheat varieties (*Rasheed & Xia, 2019*). Given the rising threat of drought, breeding bread wheat varieties with inherent drought tolerance is essential. Maintaining and exploiting the vast genetic diversity within wheat germplasm is critical to achieve this goal. This approach will ensure sustained wheat production and global food security.

Recent advancements in molecular biology have resulted in the developing DNA markers, like inter simple sequence repeat (ISSR), which offer valuable tools for investigating genetic diversity within crop germplasm collections (*Abdelghaffar et al., 2023*; *Al-Khayri et al., 2023*; *Al-Khayri et al., 2022*). ISSRs target regions flanking short microsatellites, tandem repeats of DNA sequences situated nearby and oriented in opposite directions. Amplification of these flanking regions is achieved through PCR (polymerase chain reaction) using either a single primer or a set of primers. The primer design incorporates SSR motifs anchored at the 5′or 3′end, typically consisting of 1–4 pyrimidine or purine residues (*Bornet & Branchard, 2001*). Moreover, Start Codon Targeted (SCoT) markers offer a reproducible and dominant approach for genetic analysis. SCoT employs a single 18-mer primer targeting the conserved sequence flanking the ATG translation start codon in plant genes. This method necessitates an annealing temperature as low as 50 °C (*Collard & Mackill, 2009*). Both ISSR and SCoT polymorphisms have proven valuable in characterizing cultivars, differentiating genetic resources, and introducing marker-assisted selection in various plant species (*Abdelghaffar et al., 2023*; *Al-Ghamedi et al., 2023*; *Al-Khayri et al., 2023*; *Atsbeha et al., 2023*; *Essa et al., 2023*; *Golkar & Nourbakhsh, 2019*; *Gupta, Balyan & Gahlaut, 2017*).

Genetic diversity plays a crucial role in enhancing drought tolerance in wheat, providing numerous traits that can contribute to improve crop resilience (*Megahed et al., 2022*). The abundance of genetic diversity within wheat populations allows for various adaptive responses to water stress, including variations in root architecture, water use efficiency, and drought-responsive metabolic pathways (*Rasool et al., 2022*). This genetic variability is essential for developing cultivars with improved tolerance to drought, as it enables the selection and breeding of plants that can withstand fluctuating water availability. By leveraging this diversity, breeders can enhance the ability of wheat to maintain yield and quality under adverse conditions, ultimately contributing to more sustainable and resilient agricultural systems in the face of increasing climate variability (*Gharib et al., 2021*).

This study aimed to assess the genetic diversity of 15 wheat genotypes, comprising five parental lines and their ten derived crosses, through the integration of ISSR and SCoT molecular markers with agro-morphological traits. Additionally, the study explored the combining ability of these genotypes under both normal and drought stress conditions. By elucidating the genetic relationships among these genotypes, this research seeks to identify promising lines with superior genetic makeup for developing wheat cultivars resilient to diverse environmental conditions. The comprehensive assessment of both molecular and phenotypic variation will facilitate the selection of superior genotypes for efficient breeding programs.

## MATERIALS AND METHODS

### Plant materials and experimental treatment

Five wheat genotypes were utilized in this study (Table 1). The parents in this study were selected based on diversity in drought tolerance from a preliminary screening trial. A half-diallel mating design (5×5) produced 10 F1 hybrids during the winter season

**Table 1  Pedigree and origin of the wheat parental genotypes.**

| Code | Genotype | Pedigree |
|------|----------|----------|
| P1 | Orabi-52 | New promising mutant line G-168-3-1 of M7 generation by using EMS 0.5% (Giza168-EMS), DUS no 269, 2018 year |
| P2 | Orabi-73 | promising mutant-line of M7 generation by using Gamma rays 300-Gy dose (Seds12), DUS no 270, 2018 year |
| P3 | Gemmiza 11 | BOW-s/KVZ/7C-SERI 82/3-GIZA 168-SAKHA 61 |
| P4 | Orabi-56 | New promising mutant line G-168-5-1atM7 generation by using EMS 0.5% (Giza 168-EMS), DUS no, 2023 |
| P5 | Orabi-1881 | New promising mutant line of M7 generation by using EMS 0.25% (Sakha93-EMS), DUS no 284, 2018 |

of 2020–2021. The genotypes of the parents and their offspring were assessed in field conditions at Experimental Farm of Faculty of Agriculture belongs to Zagazig University, Egypt (30°35′15″N, 31°30′07″E, 16 m asl) under ordinary growing conditions during the growing season of 2021 to 2022. The experimental site has an arid climate and receives low precipitation with an average annual rainfall of approximately 55 mm. The experiment was carried out in three replicates using a completely randomized design. The assessed genotypes (parents and F1 crosses) were represented by fifteen seeds planted in pots containing 10 kg of soil. After 15 days, the number of plants per pot was reduced to ten through thinning. Phosphorus and potassium fertilizers were applied as basal doses with a rate of 30 mg $P_2O_5$ per kg of soil for superphosphate and 50 mg $K_2O$ per kg of soil for potassium sulfate. Nitrogen fertilizer was applied in three installments at a rate of 80 mg N per kg of soil using ammonium sulfate. These installments were done at 20, 35, and 50 days after sowing, along with irrigation water. Intercultural practices such as weeding were performed as needed to maintain optimal growing conditions. To induce drought stress, the irrigation schedule for the pots was adjusted. The stressed pots received water once a week, while the control well-watered pots were irrigated every three days. Soil water tension was measured using a tensiometer to maintain appropriate irrigation levels for both the well-watered and stressed pots were maintained.

### Extraction of genomic DNA

For genomic DNA extraction, 100 grams of young wheat leaves were collected from 20 days old seedlings were employed for the extraction of genomic DNA utilizing a modified CTAB-based protocol (*Doyle, 1991*; *Scobeyeva et al., 2018*). The quantity and purity of the extracted DNA were assessed using a NanoDroP2000 spectrophotometer (Thermo Fisher Scientific, Waltham, MA, USA). The DNA concentration was adjusted to 50 ng/μL, and the isolated DNA was stored at −20 °C for subsequent amplification procedures.

### Inter-Simple Sequence Repeats (ISSR-PCR)

Genetic polymorphism analysis of wheat cultivars and their F1 hybrids was conducted utilizing ISSR-PCR. Primers for the analysis are presented in Table 2. The PCR protocol followed the methodology established by *Moreno, Martín & Ortiz (1998)*. Each reaction mixture, with a volume of 25 μL, contained the following components: 2 μL of 5x reaction
**Table 2** Characterization of ISSR and SCoT primers.

| Primer | Nucleotide sequences (5′-3′) | Tm (°C) | Molecular weight (g mol$^{-1}$) | Primer Length (bp) | GC content (%) |
|---|---|---|---|---|---|
| ISSR1 | AGAGAGAGAGAGAGAGYC | 56.3 | 5,366.6 | 18 | 52.94% |
| ISSR2 | CTCTCTCTCTCTCTCAT | 53.5 | 4,998.3 | 17 | 52.94% |
| ISSR3 | GAGAGAGAGAGAGAGATT | 54.3 | 5,685.8 | 18 | 44.44% |
| ISSR4 | AGAGAGAGAGAGAGAGC | 56.3 | 5,366.6 | 17 | 52.94% |
| ISSR5 | GAGAGAGAGAGAGAGC | 54.1 | 5,053.4 | 16 | 56.25% |
| ISSR6 | ACACACACACACACACG | 60.6 | 5,086.4 | 17 | 52.94% |
| SCoT1 | ACGACATGGCGACCACGC | 68.2 | 5,478.6 | 18 | 66.67% |
| SCoT2 | CCATGGCTACCACCGCAG | 65.8 | 5,429.6 | 18 | 66.67% |

**Notes.**
Y, C or T.

buffer, 20 ng/µL of template DNA, µL of 200 µM dNTPs, 2 µL of 25 mM MgCl2, 22 µL of primer (10 pmol), and 1 unit of Taq DNA polymerase (Promega). The thermocycling protocol commenced with an initial denaturation step at 94 °C for 5 min, followed by 35 amplification cycles. Each cycle comprised denaturation at 94 °C for 1 min, annealing at a primer-specify temperature for 1 min and extension at 72 °C for 1 min. The procedure concluded with a final extension at 72 °C for 5 min.

## Start Codon Targeted amplification

A 25 µL PCR amplification was conducted utilizing a SCoT-PCR based marker system. The reaction mixture consisted of ten µL of GoTaq Green-Master Mix, one µL of template DNA, one µL of primers, and nuclease free water to achieve a final volume of 25 µL. Thermal cycling was carried out using an Applied-Biosystems thermal cycler with the following protocol: initial denaturation at 94 °C for 5 mins, followed by 30 cycles of denaturation at 94 °C for 1 min, annealing at 50 °C for 1 min, and extension at 72 °C for 1 min.

## Gel electrophoresis

The amplified products from ISSR and SCoT reactions were separated on 1% agarose gels and visualized using ethidium bromide (MP Biomedicals, Goddard Irvine, CA, USA) staining in TBE buffer (pH 8.5). DNA fragment sizes were estimated using a 1 kbp DNA ladder.

## Agro-morphological characterization

After 60 days from cultivation, the following measurements were taken; shoot length (cm), root length (cm), shoot fresh weight (g), shoot dry weight (g), root fresh weight (g), and root dry weight (g). The proline content in the plant samples was assessed as follows: 0.5 g of leaves were ground and mixed with 10 mL of 3% aqueous sulfosalicylic-acid to create an extract. After filtration through filter paper, two mL of this extract were combined with two mL of acid ninhydrin-reagent and two mL of glacial-acetic acid. The mixture was heated at 100 °C for 1 h, followed by rapid cooling on ice. To extract the proline, 4 mL of toluene

were added to the reaction mixture, and the resulting supernatant was used for proline determination. Absorbance was measured at 520-nm employing a spectrophotometer with toluene used as the blank (*Bates, Waldren & Teare, 1973*). Additionally, following the experiment, the following traits were collected when the plants reached physiological maturity after about 140 days after sowing: the number of spikelets/spike, spike length (cm), number of grains/spike, and grain weight/spike were recorded from five main spikes from each pot.

## Data analysis

Using molecular markers, this study explored the genetic diversity and relatedness among the studied wheat genotypes and crosses. Specific PCR loci based on SCoT and ISSR techniques were employed. Each locus was classified as either absent (0) or present (1) and all loci were regarded as independent variables. Genetic diversity was assessed by analyzing the banding patterns generated from the PCR amplifications across all genotypes. The polymorphism level, a measure of genetic variation, was determined by dividing the number of loci exhibiting polymorphism (different banding patterns) by the total number of scored loci. Genetic similarities among the wheat cultivars and hybrids were computed using Dice's coefficient (*Dice, 1945*). This coefficient was determined utilizing SPSS software version 29.0.10 (*Norušis, 1993*). A clustering analysis was subsequently performed to generate a dendrogram depicting the phylogenetic relationships among the genotypes (*Rokach & Maimon, 2005*). The dendrogram, principal component, and heatmap analyses were applied R programming version 4.1.1 (*R Core Team, 2021*). The analysis of variance (ANOVA) was performed on all studied traits using a completely randomized design. Combining abilities were evaluated employing Griffing Method 4, Model 1 (*Griffing, 1956*) using R programming version 4.1.1 (*R Core Team, 2021*). Statistically significant differences among the evaluated wheat genotypes were determined employing least significant difference (LSD) test at $P < 0.01$.

## RESULTS

### Molecular analyses

The genetic diversity analysis among the developed crosses and their parental genotypes was assessed *via* ISSR and SCoT molecular markers using six ISSR primers and two SCoT primers (Fig. 1). Seventy-six loci were detected using ISSR and SCoT-PCR primers screened in the 15 genotypes (Table 3). The amplified loci/primer were 9.5. Among 76 ISSR and SCoT-PCR loci, 28 were polymorphic (9.5/primer), and 48 were monomorphic (6/primer). Polymorphism ranged from 58.3% (ISSR3) to 23% (SCoT2), averaging 36.36%. The lowest genetic distance (1.41) was observed between P1×P4 *vs.* P4×P5, as well as P3×P5 *vs* P4×P5. This suggests a close genetic similarity between these populations. Conversely, the highest genetic distance (3.61) was detected between P2×P4 *vs* P2×P5, indicating greater genetic divergence (Table 4). The Dice coefficient was employed to analyze similarity matrices constructed from data obtained with eight primers. According to Table 5, the highest similarity (0.975) was observed between P4×P5 and P1×P4, whereas the lowest similarity (0.818) was found between P2×P5 and P2. These findings may be useful for understanding

**Table 3  Number of bands (NB), monomorphic bands (MB) and polymorphic bands (PB) generated by eight primers (six ISSR and two SCoT) in 15 wheat genotypes and the related polymorphism (%).**

| Primers | NB | MB | PB | Polymorphism (%) |
|---|---|---|---|---|
| ISSR1 | 6 | 4 | 2 | 33.3% |
| ISSR2 | 8 | 5 | 3 | 37.5% |
| ISSR3 | 12 | 5 | 7 | 58.3% |
| ISSR4 | 9 | 5 | 4 | 44.4% |
| ISSR5 | 9 | 5 | 4 | 44.4% |
| ISSR6 | 10 | 8 | 2 | 20% |
| SCoT1 | 10 | 7 | 3 | 30% |
| SCoT2 | 12 | 9 | 3 | 23% |
| Total | 76 | 48 | 28 | |
| Average | 9.5 | 6 | 3.5 | 36.36% |

the genetic relationships between different wheat populations and informing breeding programs.

## Phylogeny analysis

The clustering analysis based on ISSR and SCoT banding profiles grouped the evaluated wheat genotypes into five groups A–E (Fig. 2). Cluster A included only P2×P5, while B contained P1, and C comprised P2. Besides, Group D contained four genotypes P1×P2, P1×P3, P1×P5, and P3×P4. Finally, cluster E comprised eight genotypes P4, P3, P2×P3, P2×P4, P5, P1×P4, P3×P5, and P4×P5.

## Diallel analysis

The analysis of variance exhibited significant differences among genotypes, parents, F1 crosses, and parent *vs* cross for all evaluated traits under both conditions (Table 6). Dividing the genotypic effect into GCA and SCA components showed that the mean squares of GCA and SCA were highly significant for all studied traits. The ratio of GCA/SCA was more than the unity for all evaluated traits, except root dry weight under normal conditions indicating the preponderance of additive gene effects in controlling the inheritance of these traits.

The parental genotypes P1 and P3 exhibited positive and significant general combining ability (GCA) effects for shoot fresh weight, shoot dry weight, root fresh weight, root dry weight, shoot length, and root length under well-watered conditions (Table 7). Additionally, P3 and P4 presented the highest GCA effects for number of spikelets per spike, number of grains per spike, and grain weight per spike under normal conditions. While, under drought stress, P3 and P5 had the highest GCA estimates. In contrast, P5 presented the maximum GCA effects and proved to be the best combiner under drought stress conditions.

The SCA values for the cross combinations are presented in Table 8. The highest significant and positive SCA effects for shoot fresh weight were observed in P1×P3 under well-watered conditions and P2×P3 under water scarcity conditions. For shoot dry weight, the highest significant and positive SCA values were exhibited by P1×P2, P1×P3, and

**Table 4** **Genetic distance among the five wheat cultivars and their F1 hybrids based on SCoT and ISSR banding profiles.**

| Genotype | P1 | P2 | P3 | P4 | P5 | P1×P2 | P1×P3 | P1×P4 | P1×P5 | P2×P3 | P2×P4 | P2×P5 | P3×P4 | P3×P5 | P4×P5 |
|---|---|---|---|---|---|---|---|---|---|---|---|---|---|---|---|
| P1 | 0.00 | | | | | | | | | | | | | | |
| P2 | 3.32 | 0.00 | | | | | | | | | | | | | |
| P3 | 3.00 | 2.00 | 0.00 | | | | | | | | | | | | |
| P4 | 2.45 | 2.65 | 1.73 | 0.00 | | | | | | | | | | | |
| P5 | 3.32 | 3.16 | 2.83 | 2.65 | 0.00 | | | | | | | | | | |
| P1×P2 | 3.16 | 3.32 | 2.65 | 2.45 | 2.24 | 0.00 | | | | | | | | | |
| P1×P3 | 2.65 | 3.16 | 2.45 | 2.24 | 2.45 | 1.73 | 0.00 | | | | | | | | |
| P1×P4 | 2.45 | 3.00 | 2.24 | 2.00 | 2.24 | 2.00 | 1.73 | 0.00 | | | | | | | |
| P1×P5 | 2.83 | 3.32 | 2.65 | 2.45 | 3.00 | 2.00 | 1.73 | 2.45 | 0.00 | | | | | | |
| P2×P3 | 3.16 | 2.65 | 1.73 | 2.45 | 2.24 | 2.00 | 2.24 | 2.00 | 2.83 | 0.00 | | | | | |
| P2×P4 | 2.83 | 2.65 | 2.65 | 2.45 | 2.24 | 2.45 | 2.65 | 2.00 | 2.83 | 2.45 | 0.00 | | | | |
| P2×P5 | 3.32 | 3.46 | 3.16 | 3.32 | 3.16 | 3.00 | 2.83 | 3.00 | 2.65 | 3.00 | 3.61 | 0.00 | | | |
| P3×P4 | 2.65 | 3.16 | 2.45 | 2.24 | 2.45 | 2.24 | 2.00 | 2.24 | 1.73 | 2.65 | 2.65 | 2.45 | 0.00 | | |
| P3×P5 | 2.83 | 2.65 | 1.73 | 2.00 | 2.24 | 2.45 | 2.24 | 2.00 | 2.45 | 2.00 | 2.45 | 3.00 | 2.24 | 0.00 | |
| P4×P5 | 2.83 | 3.00 | 2.24 | 2.00 | 1.73 | 2.00 | 1.73 | 1.41 | 2.45 | 2.00 | 2.45 | 3.00 | 2.24 | 1.41 | 0.00 |

Ezzat et al. (2024), *PeerJ*, DOI 10.7717/peerj.18104

**Table 5  Dice measurement for similarity coefficient of the five wheat cultivars and their F1 hybrids based on SCoT and ISSR banding profiles.**

| Genotype | P1 | P2 | P3 | P4 | P5 | P1×P2 | P1×P3 | P1×P4 | P1×P5 | P2×P3 | P2×P4 | P2×P5 | P3×P4 | P3×P5 | P4×P5 |
|---|---|---|---|---|---|---|---|---|---|---|---|---|---|---|---|
| P1 | 1.00 | | | | | | | | | | | | | | |
| P2 | 0.84 | 1.00 | | | | | | | | | | | | | |
| P3 | 0.87 | 0.94 | 1.00 | | | | | | | | | | | | |
| P4 | 0.91 | 0.90 | 0.96 | 1.00 | | | | | | | | | | | |
| P5 | 0.85 | 0.87 | 0.90 | 0.91 | 1.00 | | | | | | | | | | |
| P1×P2 | 0.86 | 0.85 | 0.90 | 0.92 | 0.94 | 1.00 | | | | | | | | | |
| P1×P3 | 0.90 | 0.86 | 0.92 | 0.93 | 0.92 | 0.96 | 1.00 | | | | | | | | |
| P1×P4 | 0.92 | 0.88 | 0.93 | 0.95 | 0.94 | 0.95 | 0.96 | 1.00 | | | | | | | |
| P1×P5 | 0.88 | 0.84 | 0.90 | 0.91 | 0.88 | 0.94 | 0.96 | 0.92 | 1.00 | | | | | | |
| P2×P3 | 0.86 | 0.90 | 0.96 | 0.92 | 0.94 | 0.95 | 0.93 | 0.95 | 0.89 | 1.00 | | | | | |
| P2×P4 | 0.90 | 0.91 | 0.91 | 0.92 | 0.94 | 0.93 | 0.91 | 0.95 | 0.90 | 0.93 | 1.00 | | | | |
| P2×P5 | 0.83 | 0.82 | 0.85 | 0.84 | 0.86 | 0.87 | 0.88 | 0.87 | 0.89 | 0.87 | 0.82 | 1.00 | | | |
| P3×P4 | 0.90 | 0.86 | 0.91 | 0.93 | 0.92 | 0.93 | 0.94 | 0.93 | 0.96 | 0.90 | 0.91 | 0.91 | 1.00 | | |
| P3×P5 | 0.89 | 0.90 | 0.96 | 0.95 | 0.94 | 0.92 | 0.93 | 0.95 | 0.92 | 0.95 | 0.93 | 0.87 | 0.93 | 1.00 | |
| P4×P5 | 0.89 | 0.88 | 0.93 | 0.95 | 0.96 | 0.95 | 0.96 | 0.98 | 0.92 | 0.95 | 0.93 | 0.87 | 0.93 | 0.97 | 1.00 |

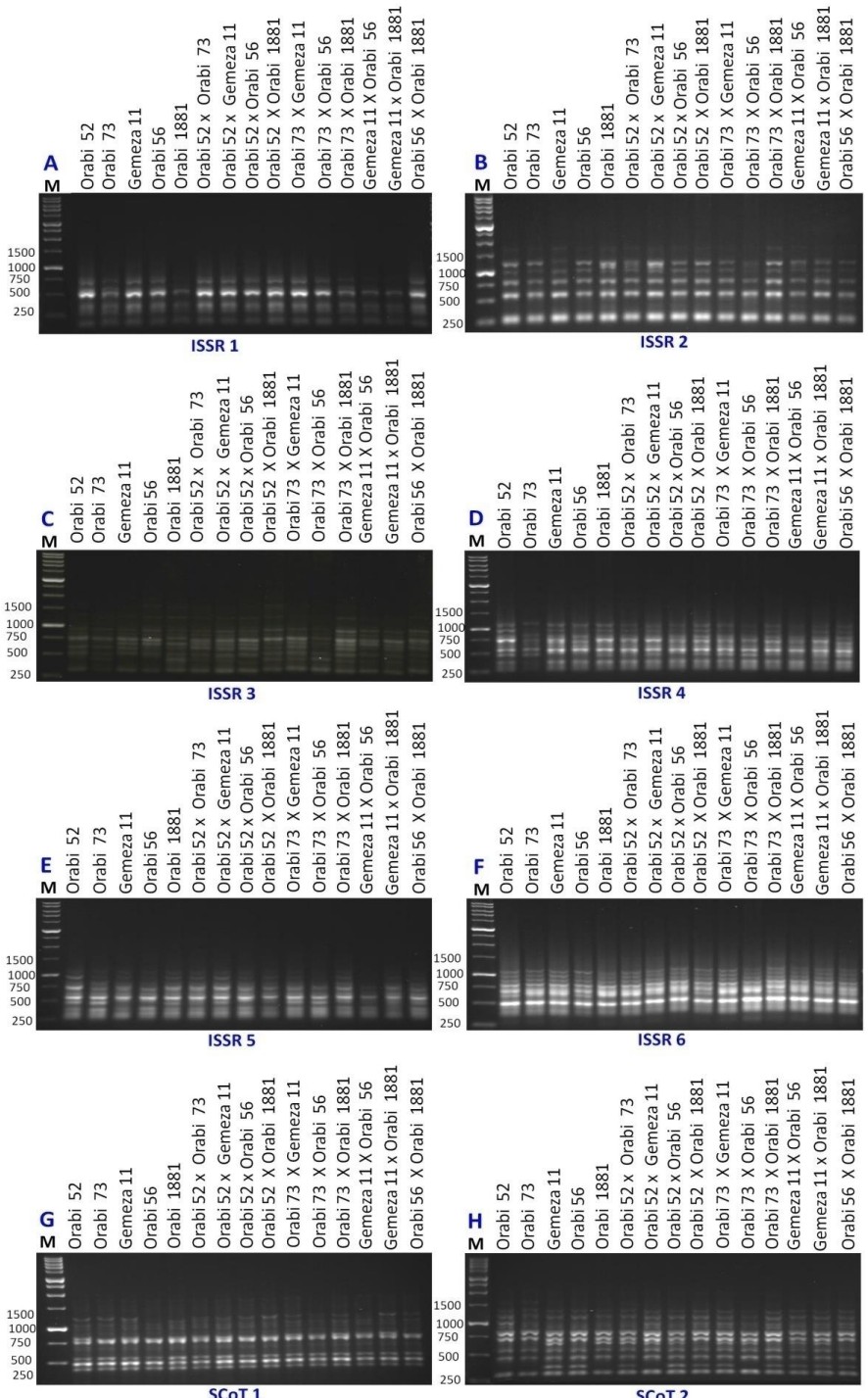

**Figure 1  ISSR and SCoT-PCR amplification patterns of 15 wheat genotypes using six ISSR primers (A–F) and two SCoT primers (G and H).** M = 1kbp DNA ladder.

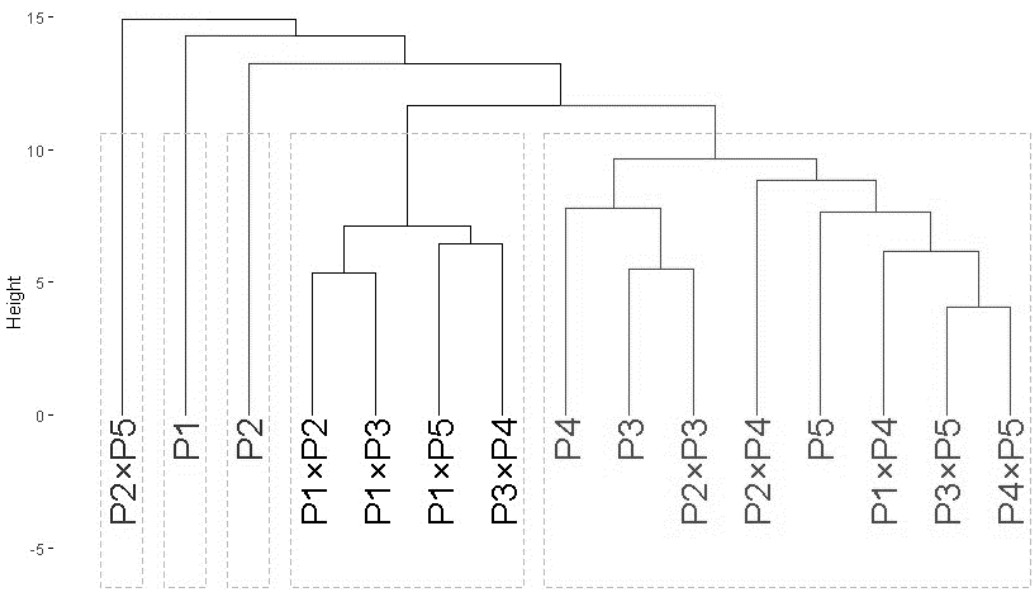

**Figure 2** **The phylogenetic tree of developed crosses and their parental wheat genotypes.** The phylogenetic tree of developed crosses and their parental wheat genotypes were revealed according to ISSR and SCoT banding profiles.

**Table 6** **Mean squares and combining for studied traits under well watered (Normal) and water deficit (Drought) conditions.**

| Source of variance | df | Shoot fresh weight | | Shoot dry weight | | Root fresh weight | | Root dry weight | | Shoot length | | Root length | |
|---|---|---|---|---|---|---|---|---|---|---|---|---|---|
| | | Normal | Drought | Normal | Drought | Normal | Drought | Normal | Drought | Normal | Drought | Normal | Drought |
| Genotype | 14 | 14.10** | 3.285** | 0.284** | 0.038** | 0.742** | 0.625** | 0.003** | 0.004** | 234.3** | 48.36** | 10.54** | 8.969** |
| Parent | 4 | 19.58** | 4.591** | 0.432** | 0.062** | 1.002** | 1.243** | 0.004** | 0.006** | 333.2** | 86.19** | 10.45** | 12.23** |
| F1 Cross | 9 | 11.41** | 3.050** | 0.218** | 0.030** | 0.563** | 0.364** | 0.002** | 0.003** | 201.8** | 35.89** | 11.17** | 8.292** |
| Parent vs. Cross | 1 | 16.38** | 0.176ns | 0.282** | 0.020** | 1.320** | 0.505** | 0.007** | 0.004** | 130.8** | 9.293** | 5.229** | 2.008** |
| GCA | 4 | 27.49** | 5.823** | 0.607** | 0.050** | 0.892** | 1.169** | 0.003** | 0.007** | 490.7** | 133.1** | 19.31** | 10.44 |
| SCA | 10 | 8.742** | 2.270** | 0.154** | 0.034** | 0.682** | 0.408** | 0.003** | 0.003** | 131.7** | 14.46** | 7.031** | 8.381** |
| Error | 28 | 0.601 | 0.489 | 0.006 | 0.0004 | 0.079 | 0.027 | 0.0003 | 0.0004 | 14.54 | 1.328 | 0.483 | 0.257 |
| GCA/SCA | | 3.145 | 2.565 | 3.936 | 1.483 | 1.308 | 2.867 | 0.893 | 2.586 | 3.727 | 9.205 | 2.746 | 1.246 |

| Source of variance | df | Proline content | | Spike length | | Number of spikelets/spike | | Number of grains/spike | | Grain weight/ spike | |
|---|---|---|---|---|---|---|---|---|---|---|---|
| | | Normal | Drought | Normal | Drought | Normal | Drought | Normal | Drought | Normal | Drought |
| Genotype | 14 | 21.06** | 5.826** | 4.953** | 4.962** | 4.467** | 5.213** | 47.78** | 35.18** | 0.207** | 0.020** |
| Parent | 4 | 50.06** | 10.12** | 6.692** | 3.475** | 5.233** | 7.442** | 49.90** | 76.43** | 0.410** | 0.048** |
| F1 Cross | 9 | 5.166* | 4.532** | 4.648** | 6.096** | 4.578** | 3.855** | 51.71** | 20.40** | 0.117** | 0.010** |
| Parent vs. Cross | 1 | 48.05** | 0.293ns | 0.747ns | 0.711ns | 0.400ns | 8.525** | 4.011ns | 3.211ns | 0.206** | 0.003ns |
| GCA | 4 | 36.57** | 15.80** | 15.04** | 12.85** | 11.30** | 10.55** | 105.8** | 67.15** | 0.565** | 0.045** |
| SCA | 10 | 14.85** | 1.837ns | 0.921** | 1.806** | 1.733ns | 3.077** | 24.59** | 22.40** | 0.064** | 0.010ns |
| Error | 28 | 2.133 | 1.346 | 0.230 | 0.208 | 0.790 | 0.414 | 2.151 | 2.394 | 0.006 | 0.001 |
| GCA/SCA | | 2.462 | 8.591 | 16.35 | 7.116 | 6.519 | 3.430 | 4.302 | 2.998 | 8.838 | 4.293 |

**Notes.**

ns, * and ** indicate non-significant, $p < 0.05$ and 0.01, respectively.
**Table 7 General combining ability estimates of the five parents for all assessed traits under well watered (Normal) and water deficit (Drought) conditions.**

| Parent | Shoot fresh weight | | Shoot dry weight | | Root fresh weight | | Root dry weight | | Shoot length | | Root length | |
|---|---|---|---|---|---|---|---|---|---|---|---|---|
| | Normal | Drought | Normal | Drought | Normal | Drought | Normal | Drought | Normal | Drought | Normal | Drought |
| P1 | $1.153^{**}$ | $-0.232^{ns}$ | $0.194^{**}$ | $-0.010^{*}$ | $0.112^{*}$ | $-0.077^{*}$ | $0.011^{**}$ | $-0.006^{**}$ | $5.897^{**}$ | $-1.304^{**}$ | $0.692^{**}$ | $-0.535^{**}$ |
| P2 | $-0.463^{**}$ | $-0.562^{**}$ | $-0.121^{**}$ | $-0.062^{**}$ | $-0.221^{**}$ | $-0.081^{*}$ | $-0.015^{**}$ | $-0.009^{**}$ | $0.408^{ns}$ | $-2.698^{**}$ | $0.427^{**}$ | $-0.715^{**}$ |
| P3 | $1.297^{**}$ | $0.494^{**}$ | $0.179^{**}$ | $0.047^{**}$ | $0.159^{**}$ | $-0.073^{*}$ | $0.006^{ns}$ | $0.002^{ns}$ | $3.379^{**}$ | $3.370^{**}$ | $0.907^{**}$ | $0.338^{**}$ |
| P4 | $-1.134^{**}$ | $-0.324^{*}$ | $-0.125^{**}$ | $-0.027^{**}$ | $-0.228^{**}$ | $-0.183^{**}$ | $-0.010^{**}$ | $-0.017^{**}$ | $-4.519^{**}$ | $-1.235^{**}$ | $-0.746^{**}$ | $-0.117^{ns}$ |
| P5 | $-0.852^{**}$ | $0.625^{**}$ | $-0.126^{**}$ | $0.052^{**}$ | $0.178^{**}$ | $0.414^{**}$ | $0.008^{*}$ | $0.030^{**}$ | $-5.166^{**}$ | $1.867^{**}$ | $-1.279^{**}$ | $1.031^{**}$ |
| LSD(gi)$_{0.05}$ | 0.310 | 0.279 | 0.031 | 0.008 | 0.112 | 0.066 | 0.007 | 0.002 | 1.524 | 0.461 | 0.278 | 0.203 |
| LSD(gi)$_{0.01}$ | 0.418 | 0.377 | 0.042 | 0.011 | 0.151 | 0.089 | 0.009 | 0.003 | 2.056 | 0.622 | 0.375 | 0.271 |

| Parent | Proline content | | Spike length | | Number of spikelets/spike | | Number of grains/spike | | Grain weight/spike | |
|---|---|---|---|---|---|---|---|---|---|---|
| | Normal | Drought | Normal | Drought | Normal | Drought | Normal | Drought | Normal | Drought |
| P1 | $-0.628^{*}$ | $-0.474^{*}$ | $-1.051^{**}$ | $-0.481^{**}$ | $-0.257^{ns}$ | $-0.891^{**}$ | $-0.267^{ns}$ | $-1.943^{**}$ | $-0.078^{**}$ | $-0.048^{**}$ |
| P2 | $-1.543^{**}$ | $1.050^{**}$ | $-0.051^{ns}$ | $-0.314^{**}$ | $-0.924^{**}$ | $-0.225^{ns}$ | $-3.362^{**}$ | $0.200^{ns}$ | $-0.046^{ns}$ | $-0.003^{ns}$ |
| P3 | $1.864^{**}$ | $-1.209^{**}$ | $1.315^{**}$ | $1.376^{**}$ | $1.029^{**}$ | $0.085^{ns}$ | $2.495^{**}$ | $-1.133^{**}$ | $0.148^{**}$ | $-0.023^{**}$ |
| P4 | $-0.425^{ns}$ | $0.415^{ns}$ | $-0.147^{ns}$ | $-0.457^{**}$ | $0.362^{*}$ | $-0.044^{ns}$ | $1.543^{**}$ | $0.105^{ns}$ | $0.185^{**}$ | $-0.001^{ns}$ |
| P5 | $0.732^{*}$ | $0.219^{ns}$ | $-0.066^{ns}$ | $-0.124^{ns}$ | $-0.210^{ns}$ | $1.075^{**}$ | $-0.410^{ns}$ | $2.771^{**}$ | $-0.208^{**}$ | $0.075^{**}$ |
| LSD(gi)$_{0.05}$ | 0.584 | 0.464 | 0.192 | 0.181 | 0.365 | 0.257 | 0.586 | 0.619 | 0.031 | 0.015 |
| LSD(gi)$_{0.01}$ | 0.788 | 0.626 | 0.259 | 0.245 | 0.480 | 0.347 | 0.791 | 0.834 | 0.040 | 0.021 |

Notes.

ns, * and ** indicate non-significant, $p < 0.05$ and 0.01, respectively.

P4×P5 under normal conditions, and by P1×P2, P1×P3, P2×P5, P3×P4, and P3×P5 under drought stress conditions. For root fresh weight, P3×P5 showed positive and significant SCA effects under drought stress. Additionally, the crosses P1×P3, P1×P4, P2×P4, P2×P5, and P3×P5 recorded the highest positive and significant SCA effects under both conditions. Regarding shoot length, the crosses P1×P2 and P1×P3 displayed significantly positive SCA effects under normal conditions, while P3×P5 did so under water-stress conditions. P4×P5 possessed the highest positive and significant SCA effects for root length under both conditions, with P2×P3 showing similar effects under drought stress. For spike length, high SCA effects were obtained by P1×P4 and P2×P3 under normal conditions, and by P2×P3, P3×P4, and P3×P5 under stressed conditions. In the case of the number of spikelets per spike, the cross P3×P5 recorded the highest positive and significant SCA effects under drought conditions. For the number of grains per spike, the cross combinations P1×P4 and P3×P5 possessed the highest values under normal conditions, while P3×P4 and P3×P5 did so under stress. Similarly, the crosses P1×P4, P1×P5, and P4×P5 were identified as good specific combiners for grain weight per spike under well-watered conditions, while P2×P3 showed the highest values under drought stress.

## Agro-morphological traits

The performance of the studied wheat genotypes and their corresponding F1 crosses for agro-morphological traits under both drought and well-watered conditions is illustrated in Figs. 3 to 5. Differences between the assessed genotypes were observed for all studied attributes. P1 and P3 exhibited high shoot fresh and dry weights under well-watered

**Table 8  Specific combining ability effects of ten F1 cross combinations for all studied traits.** Specific combining ability effects of ten F1 cross combinations for all studied traits under well watered (Normal) and water deficit (Drought) conditions.

| Cross | Shoot fresh weight | | Shoot dry weight | | Root fresh weight | | Root dry weight | | Shoot length | | Root length | |
|---|---|---|---|---|---|---|---|---|---|---|---|---|
| | Normal | Drought | Normal | Drought | Normal | Drought | Normal | Drought | Normal | Drought | Normal | Drought |
| P1×P2 | 0.145ns | 0.251ns | 0.117** | 0.035** | 0.095ns | 0.065ns | 0.009ns | 0.005ns | 3.001** | 0.741ns | 0.404ns | 0.472ns |
| P1×P3 | 0.986* | 0.237ns | 0.129** | 0.025* | 0.185ns | 0.175ns | 0.015ns | 0.010** | 3.840** | 0.832ns | 0.702ns | 0.271ns |
| P1×P4 | −1.525** | −0.687ns | −0.238** | −0.033* | −0.042ns | −0.046ns | −0.024** | 0.008* | −6.055** | 0.918ns | 0.559ns | 0.244ns |
| P1×P5 | −4.099** | −1.944** | −0.523** | −0.270** | −0.950** | −0.725** | −0.060** | −0.060** | −16.24** | −5.985** | −3.834** | −4.348** |
| P2×P3 | −0.432ns | 0.758* | −0.089* | 0.026* | −0.082ns | −0.066ns | −0.006ns | −0.017** | −1.792ns | −1.093** | 0.022ns | 1.117** |
| P2×P4 | 0.257ns | −0.190ns | 0.042ns | −0.010ns | 0.128ns | 0.145ns | 0.012ns | 0.007* | 0.297ns | −0.088ns | −0.232ns | −0.317ns |
| P2×P5 | 0.275ns | 0.336ns | −0.031ns | 0.032** | 0.193ns | 0.145ns | 0.011ns | 0.020** | 1.923ns | 0.890ns | 0.320ns | 0.202ns |
| P3×P4 | −0.286ns | −0.113ns | −0.102* | 0.035** | −0.403** | −0.151ns | −0.027** | −0.017** | −0.131ns | −0.596ns | −1.962** | −0.629* |
| P3×P5 | 0.249ns | 0.330ns | 0.050ns | 0.030** | 0.270ns | 0.246** | 0.017ns | 0.023** | 1.822ns | 0.941** | 0.562ns | 0.371ns |
| P4×P5 | 0.163ns | 0.581ns | 0.086* | −0.018ns | −0.604** | −0.537** | −0.035** | −0.044** | 1.274ns | 0.227ns | 1.049** | 1.123** |
| LSD(Sij)$_{0.05}$ | 0.400 | 0.361 | 0.040 | 0.011 | 0.145 | 0.085 | 0.009 | 0.003 | 1.268 | 0.595 | 0.359 | 0.261 |
| LSD(Sij)$_{0.01}$ | 0.540 | 0.487 | 0.054 | 0.015 | 0.196 | 0.115 | 0.012 | 0.004 | 1.955 | 0.802 | 0.484 | 0.353 |

| Cross | Proline content | | Spike length | | Number of spikelets/spike | | Number of grains/spike | | Grain weight/ spike | |
|---|---|---|---|---|---|---|---|---|---|---|
| | Normal | Drought | Normal | Drought | Normal | Drought | Normal | Drought | Normal | Drought |
| P1×P2 | −0.690ns | 0.274ns | −0.346ns | −0.183ns | 0.048ns | 0.298ns | 0.317ns | 0.921ns | −0.016ns | 0.010ns |
| P1×P3 | 0.567ns | 0.533ns | −0.213ns | −0.706** | 0.429ns | −0.678ns | 0.127ns | −0.413ns | −0.141** | 0.0001ns |
| P1×P4 | 0.316ns | −0.194ns | 0.749** | 0.127ns | 0.429ns | −0.383ns | 4.079** | 0.349ns | 0.181** | −0.006ns |
| P1×P5 | −0.961ns | −0.418ns | −0.165ns | −0.206ns | −1.667** | −1.835** | −6.302** | −4.984** | 0.120** | −0.115** |
| P2×P3 | −0.069ns | −2.053** | 1.287** | 1.627** | −0.571ns | −0.344ns | −3.111** | 0.778ns | 0.075ns | 0.072** |
| P2×P4 | 0.861ns | 0.175ns | −0.084ns | −0.706** | 0.762ns | −0.049ns | 0.508ns | −0.127ns | −0.022ns | −0.017ns |
| P2×P5 | 1.172ns | 0.331ns | −0.165ns | −0.373ns | −0.333ns | −0.168ns | 0.794ns | −0.460ns | 0.011ns | −0.021ns |
| P3×P4 | −2.336** | 0.747ns | −0.117ns | 0.603* | 0.143ns | 0.475ns | 0.317ns | 2.540* | −0.026ns | 0.030ns |
| P3×P5 | −3.690** | 0.170ns | 0.202ns | 0.603* | 0.381ns | 0.856* | 1.937* | 2.873* | 0.071ns | 0.022ns |
| P4×P5 | −2.477** | −0.135ns | −0.237ns | 0.103ns | −0.286ns | −1.249** | −0.778ns | −3.365** | 0.224** | −0.035ns |
| LSD(Sij)$_{0.05}$ | 0.754 | 0.599 | 0.248 | 0.234 | 0.459 | 0.332 | 0.757 | 0.799 | 0.039 | 0.020 |
| LSD(Sij)$_{0.01}$ | 1.017 | 0.808 | 0.334 | 0.316 | 0.619 | 0.448 | 1.021 | 1.077 | 0.053 | 0.027 |

**Notes.**

ns, * and ** indicate non-significant, $p < 0.05$ and 0.01, respectively.

conditions which was reflected in the performance of their F1 crosses P1×P3, P1×P2, P3×P5, and P2×P3 (Fig. 3). The shoot fresh weight of P3 and P1 was reduced by 40.9% and 49.2% under drought stress compared to well-watered conditions. Besides, the crosses P3×P5, P2×P3, P1×P2, and P1×P3 displayed reductions in shoot fresh weight by 24.7%, 18.9%, 41.3%, and 45.4%, respectively. Similarly, P3×P5, P2×P3, P1×P2, P1×P3, P1, and P3 exhibited reductions in their shoot dry weight by 25.1%, 26.1%, 35.9%, 37.2%, 39.6%, and 41.3%, respectively. The greatest root fresh and dry weights under water deficit were achieved by P5 (Orabi-1881) and its F1 crosses P2×P5 and P3×P5 (Fig. 3), highlighting the significance of these crosses in breeding programs. The root fresh weight of P5, P2×P5, and P3×P5 was reduced by 16.2%, 21.1%, and 27.9%, respectively, under water deficit conditions compared to well-watered conditions. However, these genotypes showed an increase in root dry weight by 3.9%, 3.5%, and 1.4%, respectively.

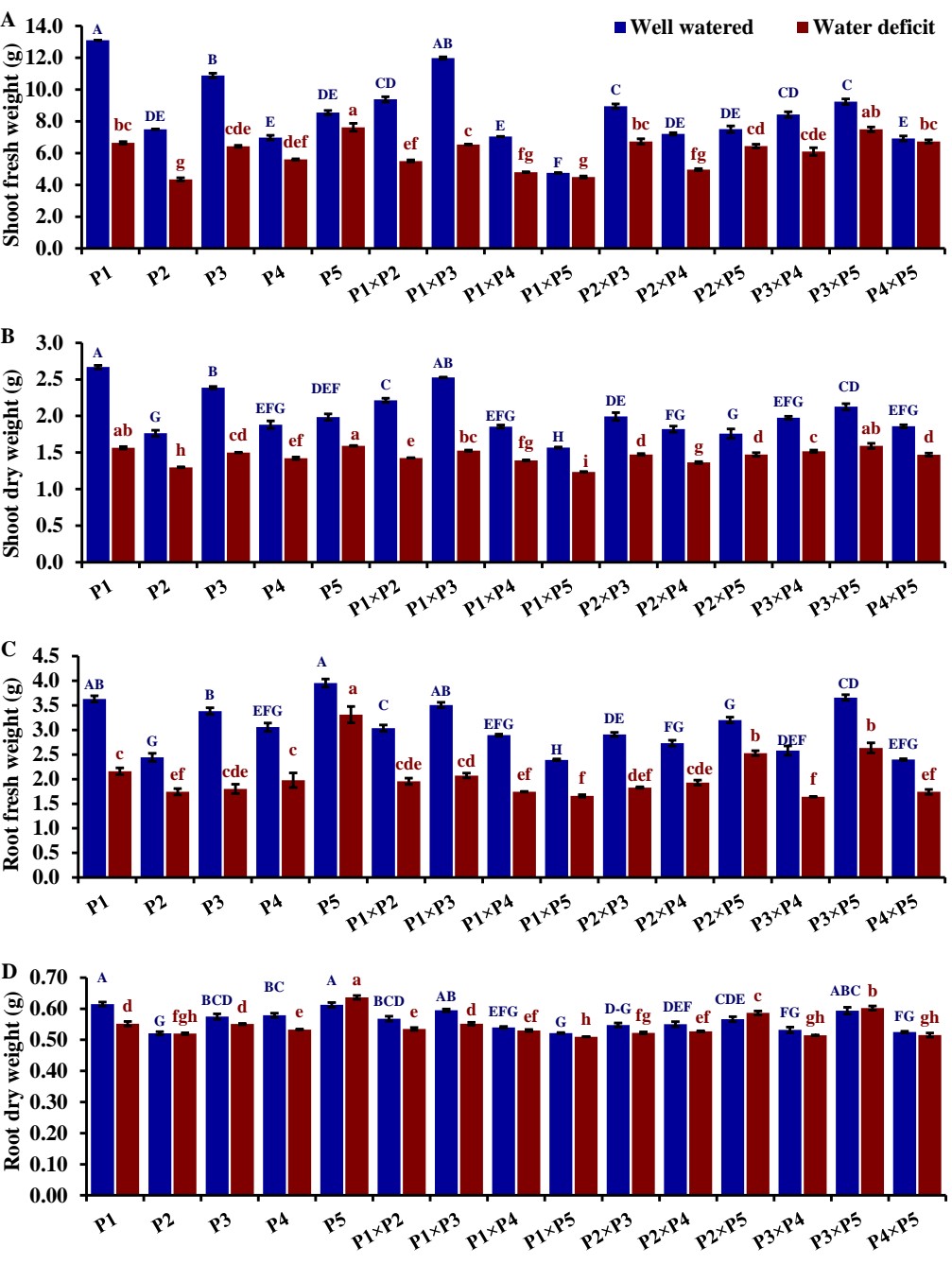

**Figure 3** **Comparative performance of developed crosses and their parental genotypes.** (A) Shoot fresh weight, (B) shoot dry weight, (C) root fresh weight, and (D) root dry weight (D). The bars at the top of the columns indicate the standard error (SE). Different letters on the columns indicate a significant difference using LSD, $p < 0.01$. Uppercase letters represent well-watered conditions, while lowercase letters represent water deficit conditions.

Drought significantly reduces overall wheat growth, which is evident in the substantial reduction in plant height for most genotypes. P1, P3, and their cross P1×P3 showed

high shoot length under normal conditions (Fig. 4). Under water deficit conditions, P3, P5, P3×P5, and P1×P3 performed best for shoot length. Under water deficit conditions compared to well-watered conditions, the genotypes P5, P3×P5, P3, P1×P3, and P1, displayed reductions in shoot length by 18.6%, 25.1%, 30.6%, 45.5%, and 55.2%, respectively. The genotypes P5 and P2×P5 maintained shoot and root length under both conditions. Root length values of P5, P4×P5, P3×P5, and P3×P4 were higher under drought than under well-watered conditions (Fig. 4). The genotypes P5, P4×P5, P3×P5, and P3×P4, displayed increases in root length by 48.9%, 21.8%, 4.0%, 3.1%, and respectively under drought stress compared to normal conditions. All genotypes showed significantly higher proline accumulation under drought stress. P2 had the highest proline content under drought and the lowest under well-watered conditions, while P3 had the opposite (Fig. 4). P2×P3, P3×P5, P3, and P3×P4 had the highest mean spike length under normal conditions, while P1×P5 and P1 had the lowest values (Fig. 5). Under drought stress compared to normal conditions, the genotypes P3×P5, P3×P4, P2×P3, and P3, exhibited reductions in spike length by 19.5%, 19.6%, 20.0%, and 25.3%, respectively. On the other hand, under drought conditions, P1 spike length was less affected compared to P2, P4, and P2×P4. P1 exhibited a reduction in spike length by 11.9%, while P2, P4, and P2×P4 showed reductions of 32.9%, 34.6%, and 41.5%, respectively under drought stress compared to normal conditions. P3×P5 possessed the uppermost number of spikelets per spike under both conditions, while P2 had the lowest number. Under well-watered conditions, P3×P4, P1×P4, and P3 showed the highest grain number per spike. However, under drought stress, these genotypes exhibited 42.5%, 50.0%, and 57.6% reductions, respectively. In contrast, P2 and P1×P5 had the lowest grain number per spike under well-watered conditions, with reductions of 30.9% and 40.1%, respectively, under drought stress. Conversely, under drought stress, P5, P2×P5, P3×P5, and P3×P4 exhibited the greatest grain number per spike, with reduction percentages of 22.7%, 29.4%, 35.7%, and 42.5%, respectively. Moreover, under drought stress, P5, P3×P4, P2×P5, and P4×P5 had the highest grain weight per spike, with reduction percentages of 42.4%, 76.0%, 65.6%, and 73.2%, respectively. Conversely, P3, P1×P3, P1×P5, and P4 showed the lowest grain weight per spike, with reductions of 81.07%, 75.33%, 74.90%, and 74.48%, respectively, indicating their higher sensitivity to drought.

## Genotypic classification

The data obtained from agro-morphological characters were employed to illustrate the relatedness among the tested wheat genotypes based on their agronomic performance under drought stress (Fig. 6). The analysis grouped wheat genotypes into three distinct clusters (A–C). Group A included three genotypes: P5, P2×P5, and P3×P5, which exhibited the best performance under drought stress, identifying them as highly drought-tolerant. Group B consisted of six genotypes: P4×P5, P1, P1×P3, P3, P3×P4, and P2×P3, which showed intermediate tolerance to drought stress. This indicates that these genotypes possess moderate drought resilience. Group C comprised six genotypes: P1×P5, P2, P1×P4, P4, P1×P2, and P2×P4, which exhibited the lowest tolerance to drought stress. These genotypes

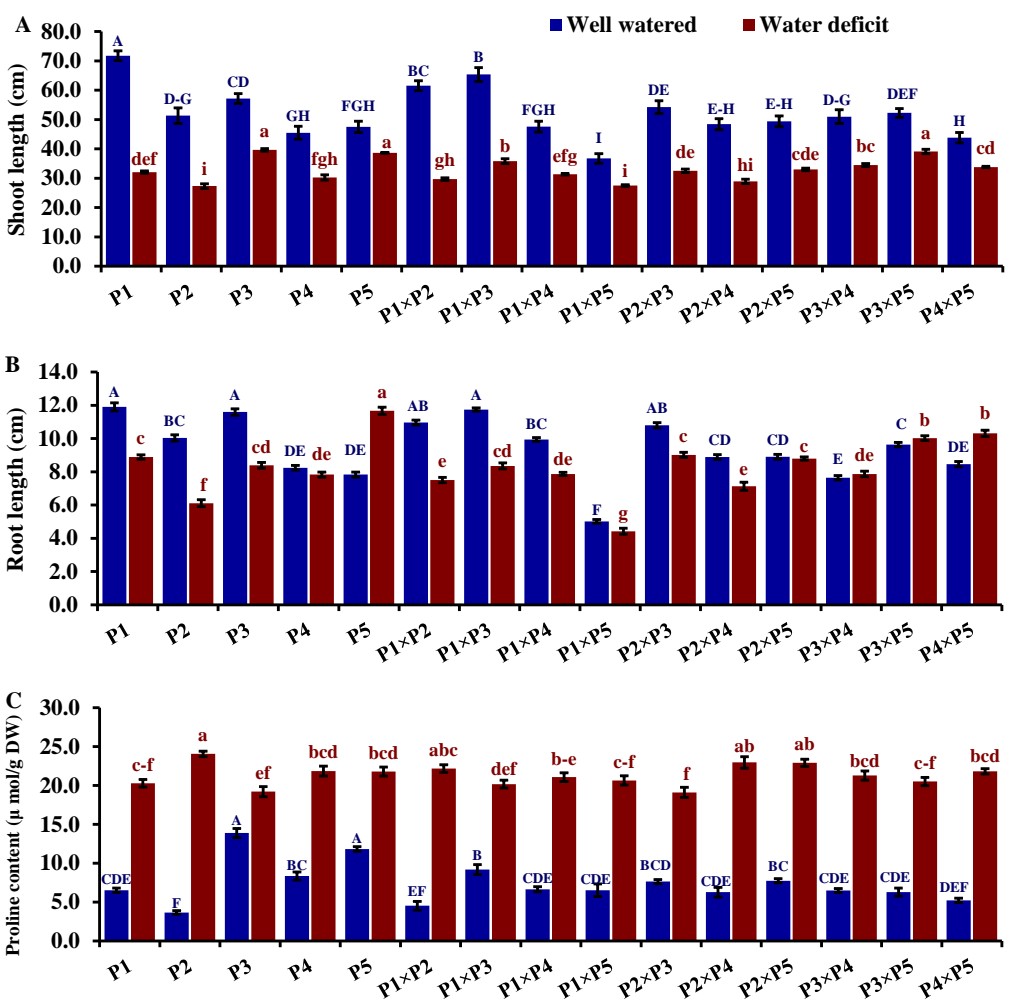

**Figure 4  Comparative performance of developed crosses and their parental genotypes.** (A) Shoot length, (B) root length, and (C) proline content. The bars at the top of the columns indicate the standard error (SE). Different letters on the columns indicate a significant difference using LSD, $p < 0.01$. Uppercase letters represent well-watered conditions, while lowercase letters represent water deficit conditions.

are considered drought-sensitive. This clustering provides valuable insights for selecting genotypes for breeding programs to improve wheat drought tolerance.

## Association among assessed genotypes and evaluated characters

Principal component analysis was performed to illustrate the association among agro-morphological attributes of the wheat crosses and their parental genotypes. The first two PCs displayed the most variance registering around 85.08% (62.36% and 22.72% for PC1 and PC2 in the same order), and were used to construct the PC-biplot (Fig. 7). PCA1 effectively categorized the assessed genotypes into groups depending on their position on the positive or negative side. The genotypes on the positive side of PCA1 were associated with high performance, particularly P5, P3×P 5, P2×P5, P3×P4, P2×P3, P4×P5, and P1.

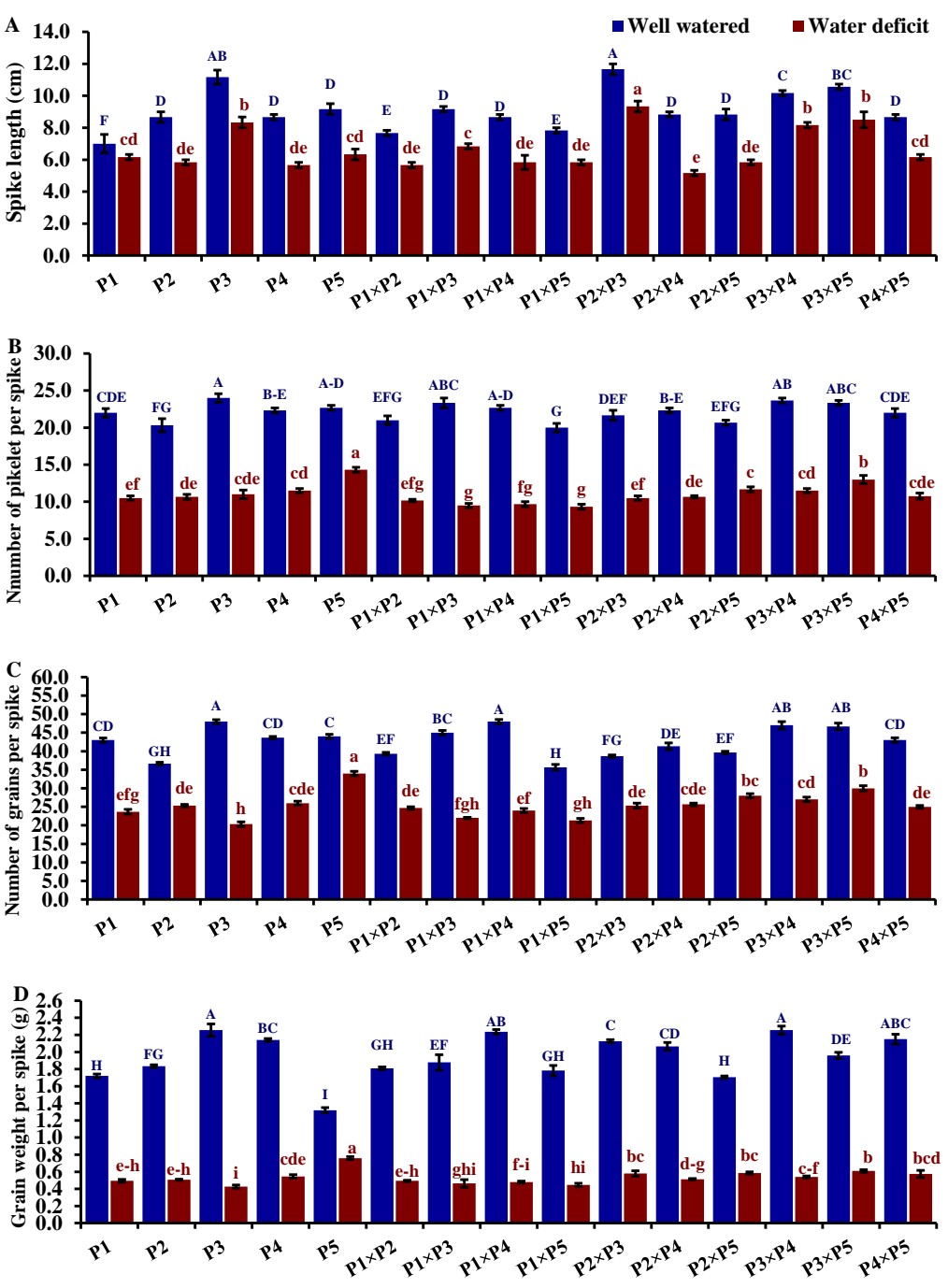

**Figure 5 Comparative performance of developed crosses and their parental genotypes.** (A) Spike length, (B) number of spikelet per spike, (C) number of grains per spike, and (D) grain weight/spike (D). The bars at the top of the columns indicate the standard error (SE). Different letters on the columns indicate a significant difference using LSD, $p < 0.01$. Uppercase letters represent well-watered conditions, while lowercase letters represent water deficit conditions.

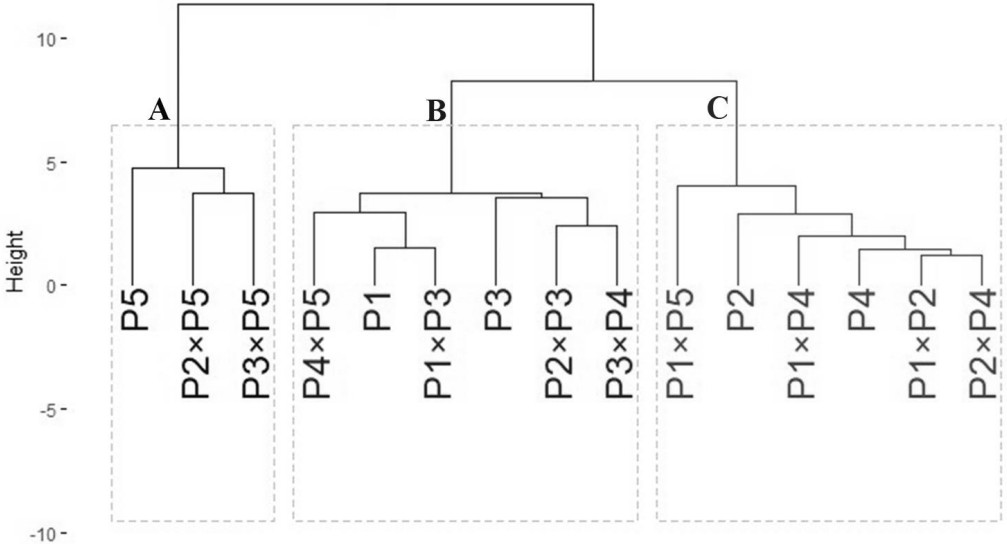

**Figure 6** Dendrogram of developed crosses and their parental wheat genotypes according to the evaluated traits under water deficit conditions.

Conversely, the genotypes on the negative side of PCA1 exhibited inferior performance, remarkably P1× P5, P2, P1×P4, P1×P2, and P2×P4. Yield-contributing traits showed a strong positive correlation with root characteristics. Moreover, heatmap based on the agro-morphological attributes characterized the genotypes into distinct groups (Fig. 8). Using a color scale under drought stress, the heatmap analysis illustrated the relationship between the assessed genotypes and the studied traits. High values of measured agronomic characteristics were displayed in blue, while low values were shown in red. The genotypes P5, P3×P5, P2×P5, P2×P3, P3×P4, and P4×P5 exhibited greater values for all agronomic attributes corresponding to blue color in the heatmap. Otherwise, genotypes P1×P5, P2, P1×P4, P1×P2, and P2×P4 had the lowest values, expressed in red under water deficit conditions.

## DISCUSSION

Genetic diversity analysis employing molecular markers and agro-morphological characterization is fundamental for wheat breeding programs to develop new stress-tolerant genotypes (*Bapela et al., 2022*). Molecular markers can facilitate this process by linking specific genetic variations to desirable traits, thereby guiding the selection of genotypes that not only exhibit high genetic diversity but also demonstrate robust performance under drought conditions. The relationship between genetic diversity and drought tolerance in wheat is fundamental to effective breeding strategies aimed at improving crop resilience under water-limited conditions. Genetic diversity within wheat germplasm provides a rich pool of alleles and traits that can be harnessed to enhance drought tolerance (*Hafeez et al., 2024*). High genetic diversity offers a broader range of adaptive traits, such as improved root traits and better water-use efficiency which are critical for surviving in drought-prone

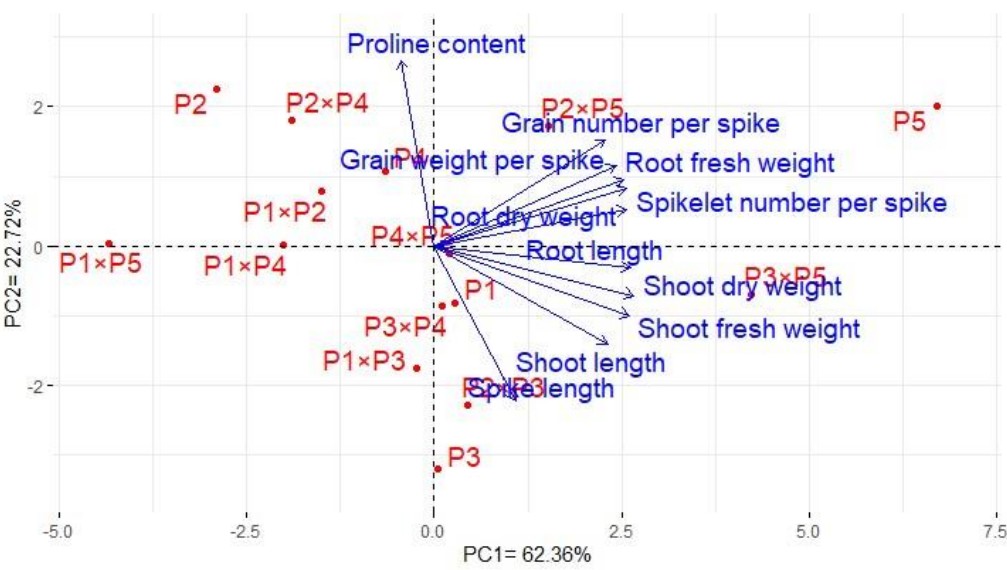

**Figure 7** The principal component biplot for the developed crosses and their parental wheat genotypes according to the traits studied under water deficit conditions.

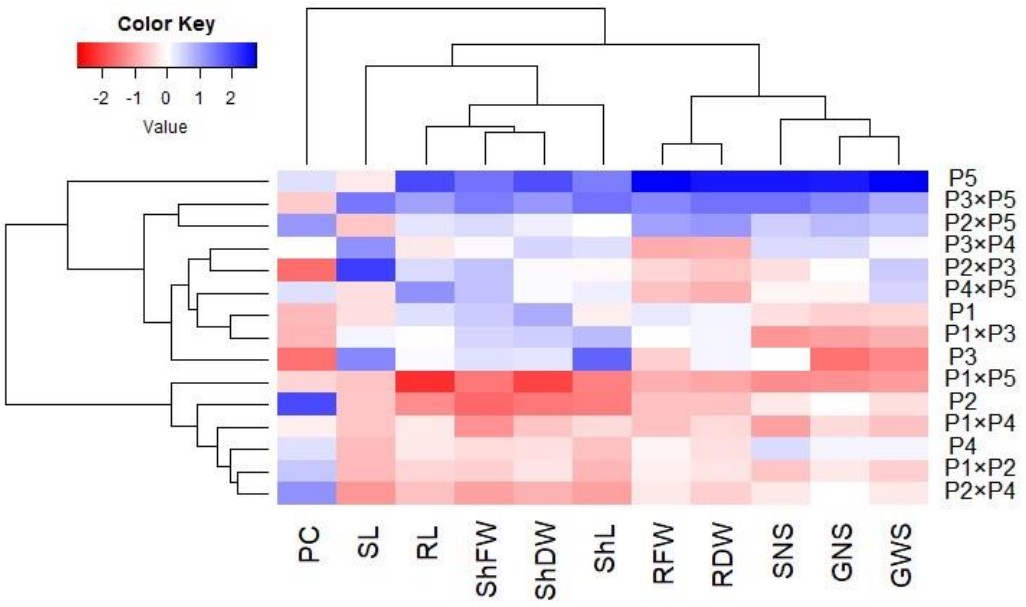

**Figure 8** Heatmap categorizing the developed crosses and their parental wheat genotypes under water deficit conditions into distinct clusters based on studied traits.

environments. By assessing genetic diversity, breeders can identify and select diverse genotypes that carry beneficial alleles associated with drought resistance. Integrating this genetic information with phenotypic traits such as root length, shoot biomass, and

yield under drought stress allows for a more comprehensive understanding of how genetic variation translates into drought tolerance. The present study demonstrated the importance of assessing molecular and agro-morphological diversity in five parental wheat genotypes and their corresponding crosses in an attempt to improve their drought resilience. Under varying conditions, the observed variation in performance differences among evaluated genotypes provided crucial insights into the genetic factors influencing these traits. This knowledge is instrumental in selecting superior genotypes for future breeding efforts, thereby enhancing the effectiveness of breeding programs.

The genetic diversity analysis utilizing ISSR and SCoT molecular markers (36.36% on average) suggested moderate genetic diversity among the wheat genotypes. The lowermost genetic distance (1.41) was detected between several cross combinations, indicating close genetic relationships. The highest genetic distance (3.61) was detected between P2×P4 and P2×P5, suggesting a more significant difference between these parental lines and their offspring. The Dice coefficient analysis revealed similar trends, with the highest similarity between P4×P5 and P1×P4 (0.975) and the lowest between P2×P5 and P2 (0.818). The genetic distances and similarity coefficients provided further insights into genotype relationships (*Herrera et al., 2021*; *Sheikh et al., 2021*).

The clustering based on ISSR and SCoT markers resulted in five clusters (A–E). This suggests that these markers may capture a broader range of genetic variations. This also suggests that ISSR and SCoT markers and may be more powerful for discriminating between closely related wheat genotypes (*Abouseada et al., 2023*; *Shaban et al., 2022*). Interestingly, P2×P5 formed a distinct cluster (A) in the ISSR/SCoT analysis, suggesting a unique genetic makeup despite its parents belonging to separate clusters (B and C). The ISSR and SCoT molecular markers employed in this study were informative and distinguished in the genetic diversity among the studied genotypes. Numerous studies have explored the molecular diversity of bread wheat using ISSR and SCoT and SSR markers (*Abouseada et al., 2023*; *Atsbeha et al., 2023*; *Jabari et al., 2023*; *Kutlu et al., 2023*; *Shaban et al., 2022*). Some studies revealed that sequence-related amplified polymorphism (SRAP) molecular marker has the great potential to determine genetic diversity (*Al-Ghamedi et al., 2023*; *Essa et al., 2023*; *Yi et al., 2021*; *Zhou et al., 2021*). Additionally, Several studies have employed SCoT markers alongside ISSR markers to assess genetic diversity in wheat germplasm. These studies include durum wheat breeding lines and landraces (*Etminan et al., 2016*), Iranian Triticum species (*Pour-Aboughadareh et al., 2017*), North African wheat cultivars (*Mohamed et al., 2017*), and *Triticum urartu* accessions (*Gholamian et al., 2019*).

Under well-watered conditions, parental genotypes P1 and P3 exhibited favorable GCA effects for several traits including shoot fresh weight, shoot dry weight, root fresh weight, root dry weight, shoot length, and root length. This implies that these genotypes contribute positively to the general performance of these traits. In contrast, under drought stress, P3 and P5 emerged as superior in terms of GCA, with P3 also showing consistent favorable GCA effects for spike length across both conditions. For reproductive traits, P3 and P4 demonstrated the highest GCA effects for the number of spikelets per spike, number of grains per spike, and grain weight per spike under normal conditions, while P5 proved to be the best combiner for these traits under drought stress. In addition,

the SCA analysis highlighted several promising cross combinations. The crosses P1×P2, P1×P3, P2×P5, P3×P4, and P3×P5 showed significant positive SCA effects for shoot dry weight under drought stress. Likewise, P3×P5 was notably effective under drought conditions improving root fresh weight. Besides, the crosses P1×P3, P1×P4, P2×P4, P2×P5, and P3×P5 demonstrated high and significant SCA effects for multiple traits under both conditions. Particularly for shoot length, P1×P2 and P1×P3 showed strong SCA effects under normal conditions, while P3×P5 excelled under water stress. The cross combinations P4×P5 and P2×P3 exhibited the highest SCA effects for root length, with P2×P3 also performing well under drought stress. For spike length, P1×P4 and P2×P3 were prominent under normal conditions, whereas P2×P3, P3×P4, and P3×P5 stood out under drought stress. P3×P5 was the most effective cross for the number of spikelets per spike under drought conditions. Additionally, P3×P4 and P3×P5 had the highest number of grains per spike under drought stress. For grain weight per spike, P2×P3 was identified as a good combiner under drought stress. In the context of identification good general combiners with high GCA or specific crosses with high SCA effects for agronomic traits under drought stress provide valuable insights for developing drought-resistant wheat varieties as indicated by *Ahmed et al. (2024)*; *Kamara et al. (2022)*; *Saeed et al. (2024)*; *Thungo, Shimelis & Mashilo (2022)*.

Considerable differences were detected between the parental genotypes and their crosses for all evaluated agro-morphological attributes. Under drought stress, the genotypes P5, P3× P5, P2×P5, P2× P3, P3×P4, and P 4× P5 exhibited superior performance, with enhanced shoot and root growth, underscoring their resilience. These genotypes appear to have inherited drought-tolerant traits, making them vital for breeding programs to improve root and shoot traits under water deficit conditions (*Zhang et al., 2017*). Moreover, the spike traits of these genotypes were also less affected by water-limited conditions compared to other genotypes. This highlights their potential to enhance drought tolerance in wheat breeding programs (*Adel & Carels, 2023*). All studied genotypes exhibited significantly higher proline accumulation under drought stress, an indicator of stress tolerance, suggesting varied stress response mechanisms among the genotypes (*Guizani et al., 2023*). In contrast, under well-watered conditions, P1 and P3 showed excellent agro-morphological performance, which was also reflected in their F1 crosses P1×P3, P1× P2, P3× P5, and P2×P3. This indicates that these genotypes possess traits that are beneficial for growth in optimal conditions. These findings underscore the importance of specific genotypes and their crosses in breeding programs aimed at both optimal growth and drought tolerance conditions (*Lazaridi et al., 2024*).

The groups of parental genotypes and their crosses classified based on agro-morphological traits and those identified through molecular analyses revealed certain differences. This divergence can be attributed to a variety of factors. Agro-morphological traits reflect the overall phenotypic performance of genotypes capturing a wide range of physiological and developmental responses (*Salem et al., 2020*; *Zannat et al., 2023*). In contrast, molecular analyses provide insights into genetic variation that may not directly correlate with phenotypic performance due to the complex interactions between multiple genes and environmental factors (*Kamara et al., 2021*; *Sakran et al., 2022*). Additionally,

molecular analyses often detect genetic variation at a more detailed level, potentially identifying loci with slight effects that are not always apparent in the phenotypic evaluations (*Igartua et al., 2015*; *Ponce-Molina et al., 2012*). This divergence highlights the need for a comprehensive approach that integrates both phenotypic and molecular analyses to accurately assess genotypic performance and identify the most promising candidates for drought tolerance. Combining these methodologies can improve the selection process for developing more resilient and tolerant wheat genotypes to environmental stresses.

## CONCLUSIONS

Unveiling genetic diversity through ISSR and SCoT marker analysis and agro-morphological characterization is crucial for the development of wheat genotypes with enhanced drought tolerance. Our findings indicate moderate genetic diversity within the studied genotypes, with distinct genetic profiles observed in specific crosses such as P2×P5. The analysis of variance revealed significant differences among assessed parental genotypes ad their corresponding F1 crosses for all evaluated traits under both well-watered and drought conditions. Notably, the parental genotypes P1 and P3 demonstrated strong GCA effects for various traits under well-watered conditions, with P3 and P5 exhibiting the highest GCA estimates under drought stress. The cross combinations P1×P3 showed the most significant positive SCA effects under well-watered conditions, while multiple crosses including P3×P5, P2×P3, and P4×P5 excelled in several traits under drought stress. These results emphasized the potential of selecting specific crosses with superior combining abilities to develop wheat cultivars with enhanced performance and resilience to environmental stresses. These results underscore the significance of genotype-specific selection to simultaneously improve drought tolerance and agronomic traits. Our study emphasizes the efficacy of integrating molecular markers and phenotypic data for the development of robust wheat breeding strategies to address the challenges of climate change and ensure sustainable crop production.

### Funding
This work was supported by Princess Nourah bint Abdulrahman University Researchers Supporting Project number (PNURSP2024R356), Princess Nourah bint Abdulrahman University, Riyadh, Saudi Arabia. The funders had no role in study design, data collection and analysis, decision to publish, or preparation of the manuscript.

### Grant Disclosures
The following grant information was disclosed by the authors:
Princess Nourah bint Abdulrahman University Researchers Supporting: PNURSP2024R356.

### Competing Interests
Diaa Abd El-Moneim and Elsayed Mansour are Academic Editors for PeerJ.

## Author Contributions

- Mohamed A. Ezzat performed the experiments, analyzed the data, prepared figures and/or tables, authored or reviewed drafts of the article, and approved the final draft.
- Nahaa M. Alotaibi analyzed the data, prepared figures and/or tables, authored or reviewed drafts of the article, and approved the final draft.
- Said S. Soliman conceived and designed the experiments, performed the experiments, analyzed the data, prepared figures and/or tables, authored or reviewed drafts of the article, and approved the final draft.
- Mahasin Sultan conceived and designed the experiments, performed the experiments, analyzed the data, prepared figures and/or tables, authored or reviewed drafts of the article, and approved the final draft.
- Mohamed M. Kamara performed the experiments, analyzed the data, prepared figures and/or tables, authored or reviewed drafts of the article, and approved the final draft.
- Diaa Abd El-Moneim performed the experiments, analyzed the data, prepared figures and/or tables, authored or reviewed drafts of the article, and approved the final draft.
- Wessam F. Felemban analyzed the data, prepared figures and/or tables, authored or reviewed drafts of the article, and approved the final draft.
- Nora M. Al Aboud analyzed the data, prepared figures and/or tables, authored or reviewed drafts of the article, and approved the final draft.
- Maha Aljabri analyzed the data, prepared figures and/or tables, authored or reviewed drafts of the article, and approved the final draft.
- Imen Ben Abdelmalek analyzed the data, prepared figures and/or tables, authored or reviewed drafts of the article, and approved the final draft.
- Elsayed Mansour performed the experiments, analyzed the data, prepared figures and/or tables, authored or reviewed drafts of the article, and approved the final draft.
- Abdallah A. Hassanin conceived and designed the experiments, performed the experiments, analyzed the data, prepared figures and/or tables, authored or reviewed drafts of the article, and approved the final draft.

## Data Availability

The raw data is available in the Supplementary File.

## Supplemental Information

Supplemental information for this article can be found online at http://dx.doi.org/10.7717/peerj.18104#supplemental-information.

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
