# Peer review of "Molecular and agro-morphological diversity assessment of some bread wheat genotypes and their crosses for drought tolerance"

_PeerJ, doi:10.7717/peerj.18104_

## Round 0.1 · original submission · Major Revisions

-Line 20: This sentence is not very clear. Please revise it.
-Line 23: Did you make a comparison only in well-watered conditions? If so, how did you determine drought tolerance? You should also write your method in the abstract in a completely understandable way.
-Line 34: Use "drought-tolerant" instead of "drought-resistant" and do the same throughout the text.
-The abstract section is quite inadequate. It needs to be improved and more details about the content of the study should be given.
-Lines 39-42: Unnecessary sentences and not very relevant to the subject. You should state the problems of wheat caused by drought. These have already been written millions of times.
-Lines 75-81: I hope you have explained the relationship between the abundance of genetic diversity and drought tolerance throughout the text. If not, the purpose of the study did not convince me much. Before crossbreeding for drought tolerance, it would be more logical to crossbreed genotypes that you know or find to be sensitive and tolerant to drought and examine their genetic diversity and segragating rates in F2. I hope you made the crosses by selecting the parents accordingly. However, I hope you conducted the study knowing that you will see only the dominant character in the phenotype in heterozygous individuals without segragating in F1 and that the obtained variation is environmentally sourced and that you discussed your results accordingly.
-Line 144-145: Are you sure that your plants grew enough in pots to measure these traits after 60 days? Also, how many spikes did you measure? You should convince us that there are enough plants for these measurements.
-The material method section should be detailed.
-More details are needed regarding the evaluation of phenotypic traits in the data analysis section. According to which experimental design was variance analysis performed? Also, if you did crossbreeding in diallel mating design, explaining the genetic parameters by performing diallel analysis will strengthen this study even more. It would be good if you added it.
-It is not very reasonable to reveal drought tolerance by determining genetic diversity in molecular analyses. It would be better if you looked at the marker-trait relationship.
-You should also present a few numerical values ​​while explaining agro-morphological traits. Also indicate the decrease or increase rates of your genotypes in the measured traits when moving from well-watered to water scarcity conditions. These are the things that give us an idea about drought tolerance.
-The discussion section is quite insufficient. This is not just a genetic diversity study, you also aimed to suggest drought-tolerant genotypes. In order to plan a breeding program, you should first know and select your parents well. In addition to my criticisms above and the recommendations of the reviewers, in order for the study to be accepted;
1) Diallel analyses should be added to the statistical analyses and associated with the results
2) Marker-trait relationships should also be added by simply analyzing them. You can benefit from the article titled "Phenotypic and genetic diversity of doubled haploid bread wheat population and molecular validation for spike characteristics, end-use quality, and biofortification capacity" on this subject.
3) Were the same genotypes included in the groups obtained from agro-morphological traits and molecular analyses? If so, and if not, why? Add this to the discussion. Which traits are more effective in selecting drought tolerant and sensitive genotypes? Are molecular markers supportive in this selection?
-If you are doing a study on drought tolerance, determining genetic diversity in molecular analyses alone will not take you anywhere. If you claim this, you should convince us with a very strong argument.
-Also, the conclusion section should be written better.
-The English of the manuscript should be improved, the terminology should be reviewed, and some typos errors should be corrected. There are quite a few of them.
-Also, carefully check the journal's rules once again and correct your manuscript.
Although the reviewers want minor revisions, my decision is for major revisions.

**Language Note:** The Academic Editor has identified that the English language must be improved. PeerJ can provide language editing services - please contact us at [email protected] for pricing (be sure to provide your manuscript number and title). Alternatively, you should make your own arrangements to improve the language quality and provide details in your response letter. – PeerJ Staff

Reviewer 1 ·

Basic reporting

You can find it in the attached pdf

Experimental design

-

Validity of the findings

-

Additional comments

-

Annotated reviews are not available for download in order to protect the identity of reviewers who chose to remain anonymous.

·

Basic reporting

Review of: Molecular and agro- morphological diversity assessment of some bread wheat genotypes and their crosses for drought tolerance
This research article assessed the genetic diversity and drought response of 5 wheat genotypes and their corresponding F1 crosses compared to well-watered conditions. The molecular proûling was conducted utilizing ISSR and SCoT markers. cultivars.
Key Findings:
• Genetic Diversity is Key: Maintaining a variety of wheat genotypes (genetic diversity) is crucial for breeding drought-tolerant cultivars.
• Drought Response Assessed: This study investigated how different wheat varieties and their offspring (F1 crosses) responded to drought conditions compared to well-watered controls.
• Molecular Profiling: Genetic fingerprinting techniques (ISSR and SCoT markers) were used to analyze the genetic makeup of the wheat varieties.
• Impact of Drought on Traits: Parental genotypes and their F1 crosses showed significant variations in drought response across several key traits, including root development, shoot growth, proline content (stress indicator), and grain production.
• Drought Tolerance Classification: Based on their performance under drought stress, the wheat varieties were categorized into groups ranging from drought-tolerant (Group A) to drought-sensitive (Group C).
• Promising Candidates Identified: Three specific genotypes (P5, P2×P5, and P3×P5) were identified as particularly promising for breeding programs due to their superior performance under water deficit conditions.
• Markers for Breeding: The study demonstrates the potential of ISSR and SCoT markers as valuable tools for identifying drought-resistant wheat varieties in breeding programs.
Overall, this is a well-researched and informative piece that contributes to understanding genetic diversity in developing drought-tolerant wheat cultivars and suggests specific markers and candidate varieties for further breeding efforts.
However, I have a few comments/ suggestions for the authors to improve the article, that are embedded into the annotated copy.

Experimental design

Well illustrated

Validity of the findings

The study presents a valid approach to investigating drought tolerance in wheat and the potential of using genetic diversity for breeding programs.

Reviewer 3 ·

Basic reporting

Dear Editor,
Thank you for the opportunity to review the manuscript titled “Molecular and agro-morphological diversity assessment of some bread wheat genotypes and their crosses for drought tolerance”. I found the topic to be very interesting and the author's approach to be insightful and well-researched

Experimental design

-Materials and methods section
-"In this experiment, 100 grams of young wheat leaves were collected from 20 days old seedlings were employed...": This sentence is a bit awkward. Consider: "For genomic DNA extraction, 100 grams of young wheat leaves were collected from 20-day-old seedlings..."
-2.6. Agro-morphological characterization "After 60 days from cultivation, measurements were taken for...: Consider "At 60 days after planting, the following measurements were taken..."
-3.3. Agro-morphological traits: Throughout this section, there are many instances where you can rephrase to avoid repetition. For example,
Instead of "superior performance compared to well-watered conditions," you could use "increased performance under drought stress."
-"Uppermost" can be replaced with "highest" for better readability. In all manuscript

Validity of the findings

The Discussion section effectively explains the findings and their significance.

Additional comments

Dear Editor,
Thank you for the opportunity to review the manuscript titled “Molecular and agro-morphological diversity assessment of some bread wheat genotypes and their crosses for drought tolerance”. I found the topic to be very interesting and the author's approach to be insightful and well-researched
Here are a few minor suggestions for improvement
-Consider replacing "drought-resistant" with "drought-tolerant" in the keywords
-In introduction, “Wheat production faces a significant challenge in water scarcity, which is increasingly becoming a critical issue in many wheat-growing regions worldwide” This sentence can be rephrased for smoother flow to “Wheat production faces a significant challenge due to water scarcity, a critical issue in many wheat-growing regions worldwide”
-Materials and methods section
-"In this experiment, 100 grams of young wheat leaves were collected from 20 days old seedlings were employed...": This sentence is a bit awkward. Consider: "For genomic DNA extraction, 100 grams of young wheat leaves were collected from 20-day-old seedlings..."
-2.6. Agro-morphological characterization "After 60 days from cultivation, measurements were taken for...: Consider "At 60 days after planting, the following measurements were taken..."
-3.3. Agro-morphological traits: Throughout this section, there are many instances where you can rephrase to avoid repetition. For example,
Instead of "superior performance compared to well-watered conditions," you could use "increased performance under drought stress."
-"Uppermost" can be replaced with "highest" for better readability. In all manuscript
-The Discussion section effectively explains the findings and their significance.

Reviewer 4 ·

Basic reporting

The manuscript entitled " Molecular and agro- morphological diversity assessment of some bread wheat genotypes and their crosses for drought tolerance " is dealing with an important issue which is the development of new cultivars adapted to drought stress which is most likely to be more severe in the climate change context. It evaluated the diversity of a group of wheat cultivar and their crosses (F1) by using molecular and phenotypic data under water stress conditions, highlighting their resilience and suitability for drought-tolerance breeding programs.
Overall, the manuscript was well written, clearly and concisely presented with focus on the most important results.

Experimental design

In the section, “Plant materials and experimental treatment, lines 84-103 ” the two experiments (field experiment and pot experiment) should described in two separate paragraph to better understand the structure of the work. Similarly for “Agro-morphological characterization, lines 134-145”.

Validity of the findings

'no comment'

---

## Round 0.2 · Minor Revisions

The changes you made have improved your manuscript considerably, but you should also correct some minor points that were overlooked. After this, your manuscript will be accepted.
1. You should include the findings on the combining ability obtained from the diallel analysis in the abstract section. Positive and significant combining abilities are important in both parental selection and determining promising hybrids, as they indicate a contribution to the relevant trait. You should synthesize these with other findings and provide a conclusion.
2. Lines 94-100: You should state your purpose clearly here. You have made it complicated by cramming literature in between. Please edit and emphasize the focus of your study. Do not ignore that the combining ability results obtained from the diallel analysis helped you make your choice.
3. Lines 182-184: Clarify which statistical program you performed the Griffing analysis. Is it R?
4. Line 209: If you use the abbreviation "P vs C" here and only once, write it as "parent vs crosses" without abbreviating it.
5. Lines 410-422: Add literature.
6. The conclusion section needs some development. Reorganize it by paying attention to the points I have indicated in the abstract. Synthesize the results of the combining ability analysis obtained from Griffing analysis molecular and phenotypic results and make recommendations for future breeding studies for drought tolerance. For example, good parents and promising hybrids. In your response letter, briefly explain the importance of the marker-trait relationships you mentioned and how the data you obtained will be used in future studies.
7. Pay attention to the words that should be italicized throughout the text. For example, Triticum urartu

---

## Round 0.3 · accepted · Accept

After your corrections, your manuscript can be accepted.